# *MAPT* Locus in Parkinson’s Disease Patients of Ashkenazi Origin: A Stratified Analysis

**DOI:** 10.3390/genes15010046

**Published:** 2023-12-28

**Authors:** Shachar Shani, Mali Gana-Weisz, Anat Bar-Shira, Avner Thaler, Tanya Gurevich, Anat Mirelman, Nir Giladi, Roy N. Alcalay, Orly Goldstein, Avi Orr-Urtreger

**Affiliations:** 1Faculty of Medicine, Tel Aviv University, Tel Aviv 6997801, Israel; shacharshani@mail.tau.ac.il (S.S.); avnert@tlvmc.gov.il (A.T.); tanyag@tlvmc.gov.il (T.G.); anatmi@tlvmc.gov.il (A.M.); nirg@tlvmc.gov.il (N.G.); aviorr@tlvmc.gov.il (A.O.-U.); 2The Laboratory of Biomarkers and Genomics of Neurodegeneration, Neurological Institute, Tel Aviv Sourasky Medical Center, Tel Aviv 6423906, Israel; maligw@tlvmc.gov.il (M.G.-W.); anatbn@tlvmc.gov.il (A.B.-S.); royal@tlvmc.gov.il (R.N.A.); 3Movement Disorders Division, Neurological Institute, Tel Aviv Sourasky Medical Center, Tel Aviv 6423906, Israel; 4Laboratory for Early Markers of Neurodegeneration, Neurological Institute, Tel Aviv Sourasky Medical Center, Tel Aviv 6423906, Israel; 5Sagol School of Neuroscience, Tel Aviv University, Tel Aviv 6997801, Israel; 6Brain Institute, Tel Aviv Sourasky Medical Center, Tel Aviv 6423906, Israel; 7Department of Neurology, Columbia University Irving Medical Center, New York, NY 10032, USA

**Keywords:** *MAPT*, Parkinson’s disease, *GBA1*, *LRRK2*, H2 haplotype, *KANSL1*

## Abstract

**Introduction**: *MAPT* locus is associated with Parkinson’s disease (PD), which is located within a large inversion region of high linkage disequilibrium (LD). We aimed to determine whether the H2-haplotype protective effect and its effect size depends on the *GBA1* or *LRRK2* risk allele carrier status, and to further characterize genetic alterations that might contribute to its effect. **Methods**: LD analysis was performed using whole-genome sequencing data of 202 unrelated Ashkenazi Jewish (AJ) PDs. A haplotype-divergent variant was genotyped in a cohort of 1200 consecutively recruited AJ-PDs. The odd ratios were calculated using AJ-non-neuro cases from the gnomAD database as the controls in an un-stratified and a stratified manner according to the mutation carrier status, and the effect on the Age at Motor Symptom Onset (AMSO) was examined. Expression and splicing quantitative trait locus (eQTL and sQTL) analyses were carried out using brain tissues from a database. **Results**: The H2 haplotype exhibited significant association with PD protection, with a similar effect size in *GBA1* carriers, *LRRK2*-G2019S carriers, and non-carriers (OR = 0.77, 0.69, and 0.82, respectively), and there was no effect on AMSO. The LD interval was narrowed to approximately 1.2 Mb. The H2 haplotype carried potential variants in candidate genes (*MAPT* and *SPPL2C*); structural deletions and segmental duplication (*KANSL1*); and variants affecting gene expression and intron excision ratio in brain tissues (*LRRC37A/2*). **Conclusions**: Our results demonstrate that H2 is associated with PD and its protective effect is not influenced by the *GBA1/LRRK2* risk allele carrier status. This effect may be genetically complex, resulting from different levels of variations such as missense mutations in relevant genes, structural variations, epigenetic modifications, and RNA expression changes, which may operate independently or in synergy.

## 1. Introduction

Parkinson’s disease (PD) is a complex progressive neurodegenerative disease, and its etiology is still largely unknown. There is clear evidence that genetic variability affects disease risk and progression. To date, genome-wide association studies (GWASs) have identified 90 independent risk-associated variants; however, the causal genes in most loci have yet to be elucidated [1]. One of these loci is a large region on chromosome 17q21, which includes an inversion of about 900 Kb, resulting in two major haplotypes, H1 and H2 [2]. While the H1 haplotype is more prevalent, the frequency of the H2 haplotype is 0.22 in European population (non-Finnish) and is exceedingly rare in African/African American population (0.17) and Asian populations (East Asian population = 0.001, and South Asian population = 0.074) [3]. Multiple lines of evidence indicate that the H1 haplotype is associated with an increased risk for PD [4], progressive supranuclear palsy [5], and Alzheimer’s disease (AD) [6], yet the precise genomic element underlying these associations remains to be determined.

Approximately nine coding genes reside within the locus, among them is the microtubule-associated protein tau (*MAPT*) gene. Tau protein, encoded by the *MAPT* gene, is primarily localized in the axons of adult neurons. Within these axons, it binds to microtubules and promotes their stability [7]. Furthermore, Tau protein undergoes phosphorylation, which, in abnormal cases, can lead to structural modifications and promote its aggregation [8].

There have been many attempts to characterize the complex genetic structure of the *MAPT* locus. Studies have shown that the inversion region exhibits a complex arrangement of structural variations [2] and that the frequency of these variants varies significantly among different populations [9]. Therefore, it is possible that using a population-based approach will be beneficial to interrogate the haplotype’s complexity.

An advantage of PD genetic studies in the Ashkenazi Jewish population (AJ) is the higher frequency of common founder mutations in the two PD-risk genes, Leucine-Rich Repeat Kinase 2 (*LRRK2)* and Glucosylceramidase β 1 (*GBA1*) (34% of AJ-PDs compared to up to 10% of PD patients in the general population), which allows the performance of targeted and specific analyses of the mutations’ dependent stratified manner [10]. Such an analysis in PD is important, as it has been demonstrated that PD patients with *LRRK2* mutations display distinct pathology and clinical manifestations in comparison to those with *GBA1* mutations or non-carrier PD patients (NC-PDs) [11]. Furthermore, the risk alleles previously identified using GWAS by us and others [1,12] can have a different effect size in subgroups of PD. For example, we demonstrated that the *PARK16* locus, in which rs823114 had a significant association with PD protection among *LRRK2*-G2019S carriers, with a large effect size, was not protective among *GBA1* carriers, suggesting a stratified effect based on mutation carrier status [13].

Here, we aim to characterize the *MAPT* locus in the AJ population, determine whether its association with PD and its effect size depends on a patient’s carrier state (patients carrying either *GBA1* mutations or *LRRK2*-G2019S mutation, and non-carriers of these mutations), and explore potential mechanisms responsible for this effect.

## 2. Materials and Methods

### 2.1. Population

Our PD cohort included 1200 patients of AJ origin, who were consecutively recruited between 2005 and 2016 (Age at Motor Symptom Onset was 60.56 ±10.96). All patients were examined at the Movement Disorder Center at the Tel-Aviv Sourasky Medical Center and underwent an interview to ascertain AJ ancestry on both parental sides, as previously described [10,14]. In total, 235 (19.6%) were carriers of mutations in *GBA1* (*GBA1-*PD) (severe: p.L444P, c.84insG, IVS2+1G>A, and p.V394L; mild: p.R496H, p.N370S, and 370Rec; and risk alleles: p.E326K and p.T369M. *GBA1*-p.R44C was also genotyped); 145 (12.1%) were *LRRK2-*G2019S PD patients (*LRRK2-*PD); 8 (0.6%) were carriers of the *SMPD1*-L302P mutation; 25 (2.1%) were carriers of mutations in more than one gene (24 *GBA1* and *LRRK2*-G2019S carriers, and 1 *GBA1* and *SMPD1*-L302P carrier); and 787 (65.6%) did not carry any of the *GBA1*, *LRRK2*, or *SMPD1* mutations (non-carriers, NC-PDs). To confirm ethnicity and the absence of hidden relatedness, principal component analysis (PCA) and identity-by-descent analysis were performed on 591 out of 1200 AJ-PDs using the Affymetrix Genome-Wide Human SNP Array 6.0 data (the Tel-Aviv PD SNP6.0 array data were described in Vacic et al., 2014 [12]). Additional information on the 1200 AJ-PD patient cohort, divided into subgroups, is detailed in Table 1.

### 2.2. Standard Protocol Approvals, Registrations, and Patient Consents

All participants provided informed consent before their involvement in the study. The DNA samples underwent coding and analysis in an anonymous manner. The study protocol and informed consent received approval from the Institutional and National Supreme Helsinki (IRB) Committees for Genetics Studies.

### 2.3. Whole-Genome Sequencing and Quality Filters

Whole-genome sequencing (WGS) was carried out on 202 unrelated AJ-PD patients (104 *GBA1-*PDs, 32 *LRRK2*-PDs, one dual-mutation carrier (*GBA1*/*LRRK2*)*,* and 65 NC-PDs); of them, 173 were included in the cohort of 1200 PDs. Sequencing was conducted using the DNBseq technology and paired-end reads were aligned to the human reference genome GRCh38/hg38 built using the BWA tool, as previously described [15]. Variant calling was performed on the alignment data of each sample using the Genome Analysis Toolkit (GATK) [16]. Variants were extracted from the *MAPT* locus (~1.4 Mb; hg38: chr17:45,394,449-46,808,970) using the SNP & Variation Suite V.8.9.0 (Golden Helix, Inc, Bozeman, MT, USA), and those with a read depth (RD) lower than 10 or genotype quality (GQ) lower than 30 were filtered out. For linkage disequilibrium (LD) analysis, variants with a call rate < 1.0 and indels were also excluded.

### 2.4. Variant Annotations

Variants were annotated using RefSeq, and evaluated for their deleteriousness using the in silico prediction scores generated by combined annotation-dependent depletion (CADD; v1.6) [17]. Additional variant characterizations were conducted using The Encyclopedia of DNA Elements (ENCODE) to annotate candidate cis-regulatory elements (cCREs) [18] and Aminode to identify evolutionarily constrained regions (ECRs) [19].

### 2.5. Genotyping

rs17651549 in *MAPT* was genotyped using the complete cohort of 1200 AJ-PD patients (Thermo Fisher Scientific fluorescent TaqMan^®^ assay: p.Arg370Trp, C__25609347_10; StepOnePlus RT-PCR system, Applied Biosystems, Beverly, MA, USA).

### 2.6. Statistical Analysis

To identify variants unique to the H2 haplotype, LD (r^2^) was calculated between the H2-tagged single-nucleotide variation (SNV) rs8070723 and each SNV in a window of 1200 kb using PLINK1.9 (https://www.cog-genomics.org/plink/1.9/, accessed on 22 May 2023) [20]. The odds ratios (ORs) and 95% confidence intervals (CIs) were calculated using ‘MedCalc’ (https://www.medcalc.org, accessed on 22 May 2023) for allelic, dominant, and recessive models for H2. The gnomAD data set of AJ-non-neuro cases (version V2.1.1; ethnicity confirmed by both self-report and PCA analysis [21]) was used as the controls [3]. Importantly, it exclusively consists of samples from individuals who were not ascertained for having a neurological condition in a neurological case/control study. The SPSS statistics software v.25 was used to conduct a stratified linear regression analysis under a dominant model of rs17651549 with Age at Motor Symptom Onset (AMSO). We previously demonstrated that the AMSO of *GBA1-*PDs was affected by both the type of *GBA1* mutations (severe or mild) and mutation dosage [10]. To avoid a confounding effect, we excluded severe *GBA1* mutation carriers, compound heterozygotes, and N370S homozygotes from this analysis (n = 67). The regression coefficients and 95% CIs were estimated, and the analysis was adjusted for sex.

The expression quantitative trait loci (eQTLs) and splicing quantitative trait loci (sQTLs) for variants within the *MAPT* locus were obtained from the GTEx Portal (https://gtexportal.org/, accessed on 20 August 2023). As GTEx measures the effect of each SNV on eQTL/sQTL within a 2 Mb interval (1 Mb upstream and 1 Mb downstream), we used three SNVs in the LD region to assess the effect of the complete 1.2 Mb H2 haplotype: rs4528616 at the proximal end, rs62071573 at the distal end, and rs17651549 located at the center of the LD region. The total region of analysis was hg38: chr17:44,606,231-47,265,628, 2.67 Mb. We determined the significance threshold for the eQTLs based on *m*-value ≥ 0.9 and *p* < 0.0005 and for the sQTLs based on *p* < 0.0001.

## 3. Results

### 3.1. Linkage Disequilibrium Analysis

A total of 13,449 variants were extracted from 202 PDs at the position hg38- chr17:45, 394,449-46,808,970 (~1.4 Mb). After filtering (see methods), 7503 variants remained.

Using the tagging SNV rs8070723, 10 homozygous individuals were identified as H2/H2. The LD analysis comparing these 10 individuals revealed two haplotype breaking points that define a 1.2 Mb minimal LD interval (hg38:chr17:45,494,449-46,708,970; Figure 1): one H2/H2-PD carried a recombinant allele at the proximal end (PD-8, Figure 1), and two H2/H2-PDs carried a recombinant allele at the distal end (PD-9 and PD-10, Figure 1). An interrogation of the WGS reads based on a visualization of the BAM files revealed two low-coverage regions. The first region, spanning a 78.6 Kb interval (chr17:45,494,872-45,573,510, Figure 1, left shaded box), is located adjacent to the proximal end of the identified 1.2 Mb minimal LD interval. This region includes two pseudogenes, *LRRC37A4P* and *RDM1P1*, as well as two copy number variants (CNVs). The second region, spanning 409.8 Kb (chr17:46,297,396-46,707,168, Figure 1, right shaded box), is located adjacent to the distal end and encompasses four genes, *LRRC37A* and *ARL17A*, with a high homology to *LRRC37A2* and *ARL17B*.

### 3.2. H2 Haplotype Analysis

Out of the 7503 variants, 6665 were in the minimal LD interval of 1.2 Mb. The LD analysis of these variants revealed 2160 haplotype-divergent redundant SNVs (r^2^ ~ = 1) (Appendix A).

We conducted a comprehensive characterization of the haplotype-divergent SNVs predicted to be pathogenic using in silico analysis tools. Out of 2160 haplotype-divergent SNVs, 78 SNVs had a CADD Phred score of 12.37 or higher (Appendix A), placing them at the top 2% of scores among all possible changes in the human genome. This specific threshold is associated with the detection of potentially pathogenic variants [22]. Among these, five SNVs had a CADD Phred score of 20 or higher, thus positioning them within the top 1% of deleterious variants (Table 2). Three of these are missense variants, one resides within *MAPT* and two reside in Signal Peptide Peptidase-Like 2C (*SPPL2C*). All missense variants were predicted to reside within evolutionarily constrained regions (ECRs) by Aminode [19]. In addition to the missense variants, one intronic variant, which was annotated to both intron 4 of Corticotropin-Releasing Hormone Receptor 1 (*CRHR1*) and intron 6 within *LINC02210-CRHR1*, was predicted to reside within a distal enhancer-like signature, as indicated by the ENCODE Registry of candidate cis-regulatory elements [23]. Lastly, there is one variant located about 2 Kb upstream to the KAT8 Regulatory NSL Complex Subunit 1 (*KANSL1*), which resides within a promoter-like signature [23].

### 3.3. H2 Is Associated with All Subgroups of PD

Since rs17651549 had the highest CADD Phred score (24.9, Table 2, Appendix A), we genotyped it in the larger cohort of 1200 AJ-PDs. This variant showed an association with PD (allelic odds ratio (OR) = 0.793, CI = 0.705–0.891, *p* = 0.0001, Table 3), confirming previous reports. Furthermore, in a dominant genetic model for the H2 allele, rs17651549 was found to be significantly associated with PD (dominant OR = 0.748, CI = 0.650–0.862, *p* = 0.0001). However, no significant association was observed under the H2/H2 recessive model (recessive OR = 0.772, CI = 0.564–1.056, *p* = 0.106), suggesting a dominant effect of the H2 allele in PD protection. Similar associations and effect sizes were observed in the subgroups of PD patients, *GBA1*-PD, *LRRK2*-PD, and NC-PD (allelic OR = 0.774, CI = 0.615–0.975, *p* = 0.030; allelic OR = 0.694, CI = 0.515–0.936, *p* = 0.017; and allelic OR = 0.815, CI = 0.712–0.933, *p* = 0.003, respectively, Table 3), suggesting a unified protection effect regardless of the carrier status of *GBA1* and *LRRK2* mutations. This protective effect was observed in these PD subgroups under the dominant mode of inheritance as well (*GBA1*-PD: dominant OR = 0.700, CI = 0.530–0.923, *p* = 0.012; *LRRK2*-PD: dominant OR = 0.678, CI = 0.478–0.961, *p* = 0.029; and NC-PD: dominant OR = 0.771, CI = 0.655–0.909, *p* = 0.002).

To further support the stratified association, and as the number of AJ controls who carried the *GBA1* mutations or the *LRRK2*-G2019S mutation was low, we conducted a simulation using gnomAD allele frequency reports. Among the 235 *GBA1*-PDs (who did not carry the *LRRK2*-G2019S mutation or the *SMPD1*-L302P mutation), 86 also carried the H2 haplotype (36.6%), and this was significantly lower than the predicted dual-carrier rate of both *GBA1* mutations and H2 haplotype (44.7%) based on genetically matched gnomAD frequencies (OR = 0.714, CI = 0.546–0.935, *p* = 0.014 when simulating with 100,000 AJs). Among the 145 *LRRK2*-G2019S-PDs (who did not carry the *GBA1* mutations or the *SMPD1*-L302P mutation), 52 also carried the H2 haplotype (35.9%), and this was significantly lower than the predicted dual-carrier rate of both *LRRK2*-G2019S and H2 haplotype (44.7%) based on genetically matched gnomAD frequencies (OR = 0.692, CI = 0.487–0.984, *p* = 0.040 when simulating with 100,000 AJs).

### 3.4. H2 Is Not Associated with Age at Motor Symptom Onset

The stratified linear regression analysis under a dominant model of H2 revealed that rs17651549 is not associated with AMSO in *GBA1*-PDs (β: 0.770, *p* = 0.615), *LRRK2*-PDs (β: 0.028, *p* = 0.944), and NC-PDs (β: 0.056, *p* = 0.291).

### 3.5. Structural Variations in H2 Haplotype

The WGS interrogation revealed structural variations located on the H2 haplotype: (i) a duplication of about 150 Kb, which covers the 5′ coding exons of the *KANSL1* gene (Appendix A), and this duplication has been previously reported to generate a novel *KANSL1* transcript [9]; (ii) a CNV with a deletion of approximately 250 bp within intron 6 of *MAPT* (chr17:46,009,350-46,009,598); (iii) A CNV with a deletion of approximately 315 bp within intron 2 of *KANSL1* (chr17:46,099,041-46,099,354); and (iv) a CNV with a deletion of approximately 315 bp within intron 2 of *KANSL1* (chr17:46,146,545-46,146,859). A subsequent analysis of the methylation and expression data within these CNV regions was performed using the UCSC GRCh37/hg19—‘UCSF Brain DNA Methylation track’ [24]. Methylated DNA immunoprecipitation (MeDIP) signals were detected in all three CNVs. The CNV within *MAPT* displayed methylation at five CpG sites, while the CNVs within *KANSL1* (iii and iv) exhibited 23 and 24 methylated CpG sites, respectively (Appendix A). All four structural variants, including the segmental duplication and CNVs, were present in all H2 carriers in our WGS cohort.

### 3.6. The Effect of H2 Haplotype on RNA Expression and Splice Variant Expression

eQTL and sQTL analyses were performed using data from all 13 available brain tissues in the GTEx Portal (amygdala, anterior cingulate cortex, caudate, cerebellar hemisphere, cerebellum, cortex, frontal cortex, hippocampus, hypothalamus, nucleus accumbens, putamen, spinal cord (cervical c-1), and substantia nigra). The eQTL analysis revealed a significant association between H2 (represented by the three haplotype-divergent SNVs) and altered gene expression in the 13 brain tissues. Specifically, H2 exhibited consistent correlation across all 13 brain regions, resulting in increased expression of two protein-coding genes, Leucine-Rich Repeat-Containing 37 Member A (*LRRC37A*) and *LRRC37A2,* and one antisense *KANSL1-AS1.* Several pseudogenes also showed a significant increase in expression (*RP11-259G18.3*, *MAPK8IP1P2*, *MAPK8IP1P1*, *DND1P1*, and *RP11-259G18.1*), while one showed a decreased expression (*LRRC37A4P*; Appendix A).

The sQTL analysis revealed a significant association between H2 and the intron excision ratios of both *KANSL1* and *LINC02210* across all 13 brain tissues. Notably, while the impact of H2 on the *LINC02210* intron excision ratio varied across different brain tissues, there was a consistent decrease in the intron excision ratio of *KANSL1.* However, the differences in intron excision in *KANSL1* need to be interpreted cautiously due to the presence of segmental duplication in this region. Other genes showed significant differences in the intron excision ratio but not across all brain tissues (Appendix A).

## 4. Discussion

In this study, we identified a 1.2 Mb *MAPT*-H2 haplotype in the AJ population and showed its dominant PD-protective effect in AJs. Studies in the AJ-PD cohort allowed us to stratify them according to subgroups of *GBA1* and *LRRK2*-G2019S mutation carriers and, importantly, to demonstrate that this protective effect of *MAPT*-H2 is independent of *GBA1* and *LRRK2* mutation status. However, there was no association between the *MAPT*-H2 haplotype and AMSO among these different PD subgroups. It is important to note that other non-H2 divergent variants within the *MAPT* locus may have an effect on the AMSO, as we previously showed an association between a *MAPT*-SNV and AMSO in PD *LRRK2*-G2019S carriers [25], which was not an H2 divergent variant.

Determining the precise genetic alteration responsible for the H2 effect in PD has been exceedingly challenging due to the structural complexity of the *MAPT* locus, which encompasses a substantial number of variants and structural variations (duplication and deletions). In an effort to further unravel this complexity, we conducted different bioinformatic analyses to identify candidate genes and variants that might contribute to the H2 PD-protective effect. First, we used WGS data to characterize potentially pathogenic haplotype-divergent SNVs, thereby revealing candidate variants and genes. One of these candidate genes is *SPPL2C*, which consists of two missense variants out of the five in silico potentially pathogenic variants described here. *SPPL2C* belongs to the intramembrane-cleaving protease family, and its disruption has been demonstrated to impair vesicle trafficking, resulting in a defective transport process within cells [26]. Impairments in intracellular trafficking are key mechanisms underlying PD pathology [27] and involved in PD risk, as we recently showed [28], thus raising the possibility that *SPPL2C* function might be involved in PD.

Next, we conducted a fine-mapping analysis using our WGS data to further confirm the presence of unique structural variants. One of them is the partial duplication of the 5′ end of the *KANSL1* gene [9]. *KANSL1,* part of the NSL complex, acts as a scaffolding protein [29] and interacts with the WD Repeat Domain 5 (*WDR5*) gene, which has been identified as being associated with the immune function of PD [30]. Both *KANSL1* and lysine acetyltransferase 8 (*KAT8*) have been implicated in PINK1-dependent mitophagy [31], a cellular process involved in the degradation of dysfunctional mitochondria, which is associated with PD [32]. In a biological screening assay focused on PD candidate genes involved in PINK1-dependent mitophagy, knockdown of both *KANSL1* and *KAT8* resulted in reduced accumulation of phospho-ubiquitin, an initiation marker of PINK1-dependent mitophagy, suggesting *KANSL1* plays a role in mitophagy regulation and PD pathology [31,33]. Interestingly, a recent study utilizing the PPMI database revealed that individuals who carry the H2 haplotype have higher levels of *KANSL1* transcript compared to those who carry H1, with H2 acting in a dose-dependent manner [34], highlighting the association between the H2 haplotype and increased transcript levels of *KANSL1*. As the H2 haplotype has a protective effect on PD, it is plausible to suggest that the presence of this additional novel *KANSL1* transcript in H2 carriers may contribute to the observed protective effect in PD.

Additionally, we observed on the H2 haplotype two CNVs with deletions within intron 2 of *KANSL1* in regions that include multiple methylated CpG sites. Intragenic DNA methylation has a well-established role in regulating the usage of alternative promoters and influencing gene expression [24]. Given that only H2 carriers display these deletions, our data suggest that different DNA methylation patterns between the H1 and H2 haplotypes could potentially modulate *KANSL1* expression. In line with this hypothesis, it has been demonstrated that individuals with neurodegenerative diseases who carry the H1 haplotype exhibit decreased *KANSL1* expression [35]. As we demonstrated a dominant protective effect of H2 in AJ-PDs, it is tempting to suggest a hypothetical state, where an additional transcript of *KANSL1* in the H2 haplotype may contribute to a dominant gain-of-function effect, which is not affected by the presence of the H1 haplotype, and results in higher levels of *KANSL1* among PDs. Altogether, these structural variants within *KANSL1* present potential factors that could impact its expression and affect PD pathology, as previously suggested [31,33].

Of note, the WGS fine-mapping analysis reduced the LD interval in the AJ cohort to a minimal region of 1.2 Mb. However, this region might be even smaller as the proximal and distal ends of this region, containing pseudogenes and CNVs, had a low coverage due to the limitation of the WGS short-read technology (~100–150 bp in length). Re-sequencing this region using other technologies that provide long reads may contribute to a more accurate characterization of this region and may reduce the LD even more.

In addition, the results of our eQTL analysis indicate that the H2 haplotype increases the RNA expression levels of *LRRC37A* and its paralog *LRRC37A2 (LRRC37A/2)* across 13 brain tissues, consistent with prior findings [36]. *LRRC37A/2* has been linked to immune and inflammatory responses, cellular migration, and synapse formation [37,38]. Notably, recent research has demonstrated that increased expression of *LRRC37A/2* leads to the upregulation of pro-inflammatory genes, thereby mediating astroglial inflammation [39]. Additionally, it has been shown that in PD, *LRRC37A/2* interacts with α-synuclein in astrocytes of the substantia nigra [39]. As the H2 haplotype is associated with elevated *LRRC37A/2* RNA levels, this may suggest a potential role for *LRRC37A/2* in PD pathology.

The minimal 1.2 Mb interval includes the *MAPT* gene, which encodes Tau protein, known to be involved in Alzheimer’s disease (AD) and other Tauopathies. Emerging evidence suggests that Tau is involved in PD pathophysiology [40]. Yet, the link between the various differences we observed between the H2 haplotype and H1 haplotype and Tau pathology (Tau aggregation and hyper-phosphorylation) is not known and should be studied.

This study primarily focused on the AJ population, which provides an advantage in our capability to examine the effect of genetic variants in stratified groups of PDs that carry founder mutations. However, our study has a few limitations. We used data generated by the short-read sequencing technique, which resulted in two low-coverage regions adjacent to the inversion. This limitation stems from challenges posed by CNVs, pseudogenes, and genes with high homology, preventing us from accurately assessing the minimum LD region and understanding its impact on PD protection. Another limitation of this study is that due to our small sample size, we did not address modification of disease penetrance for the H1/H1 haplotype and the H2 haplotypes (H2/H2 and H1/H2) using stratified GWAS as reported by Senkevich et al. [41]. An additional limitation of this study is the absence of a fully characterized motor and cognitive clinical phenotype for all 1200 PD patients, which results in a reduced power to identify any significant association between *MAPT*-locus haplotypes and PD clinical characteristics.

Nevertheless, we demonstrated here a protective effect across all subgroups. Our analyses suggest that the protective effect of H2 could be complex, resulting from different levels of variations such as missense mutations in relevant genes, structural variations, epigenetic modifications, and RNA expression alterations. These effects may operate independently or may exhibit synergistic effects. Therefore, cellular models separating these elements, as well as models combining them, are warranted to decipher the various biological mechanisms underlying the protective effect of the *MAPT* locus in PD.

## Figures and Tables

**Figure 1 genes-15-00046-f001:**
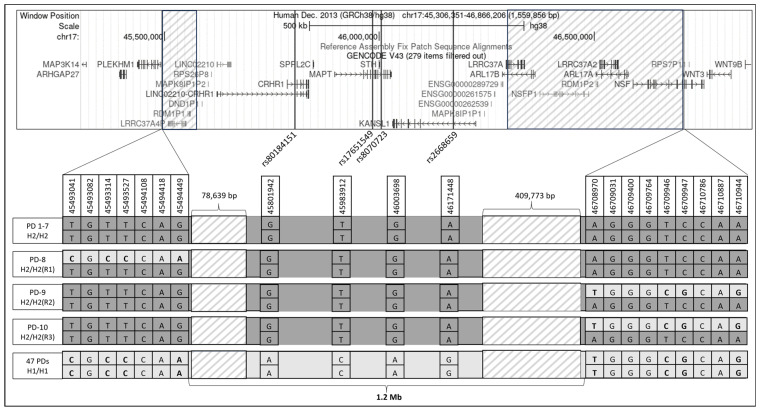
Schematic representation of the MAPT locus and the linkage disequilibrium (LD) region in Ashkenazi Jews (AJs). The upper panel presents the genes in chr17:45,306,351-46,866,206 from the hg38 UCSC genome browser GENCODE V43 annotation track. Variants are represented by black lines, while boxes filled with transparent grey diagonal lines indicate regions with low WGS coverage. Black-colored genes represent coding genes, and grey-colored genes indicate pseudogenes. In the lower panel, the specific genotypes of 10 H2/H2 AJ-PD carriers and 47 AJ-PD H1/H1 carriers are depicted schematically and not drawn to scale. Three H2/H2 individuals (PD8, PD9, and PD10) carry recombinant alleles marked as R1, R2, and R3. These three reduce the LD region to a 1.2 Mb interval.

**Table 1 genes-15-00046-t001:** The cohort of 1200 AJ-PDs divided into genetic subgroups.

Genotype	Number of PD Patients (%)	Number of Females (%)	Average Age at Onset (±SD)
Carriers of *GBA1* mutations ^a^	235 (20%)	94 (40%)	58.7 (10.5)
Carriers of *LRRK2*-G2019S mutation	145 (12.1%)	65 (44.8%)	58.5 (10.5)
Carriers of dual mutations	25 (2.1%)	17 (68%)	58.5 (9.9)
Carriers of *SMPD1*-L302P mutation	8 (0.7%)	3 (37.5%)	55.5 (12.7)
Non-Carriers (NC)	787 (65.6%)	297 (37.7%)	61.5 (11.4)
Total	1200	476 (39.6%)	60.5 (11.2)

Note: ^a^ 10 *GBA1* mutations (severe *GBA1* mutations = c.84insG, IVS2+1G>A, p.V394L, and p.L444P; mild *GBA1* mutations = p.R496H, p.N370S, and 370 Rec; and risk alleles = p.E326 K, p.T369 M; and *GBA1*-p.R44 C). SD = standard deviation.

**Table 2 genes-15-00046-t002:** Characterization of five haplotype-divergent SNVs with CADD Phred score higher or equal to 20.

Chromosome: Position (hg38)	Reference > Alternates	Identifier	CADD Phred Score	Gene Names	Sequence Ontology	Amino Acid Change	Gene Region	In Evolutionarily Constrained Region (Aminode)	ENCODE- (ENCODE Accession- cCREs)
17:46225515	C>T	rs2532404	20.70	*KANSL1*	Upstream variant	N.A	Upstream	N.A	EH38E1866648—promoter-like signature
17:45983912	C>T	rs17651549	24.90	*MAPT*	Missense variant	NP_001364194.1: p.Arg445Trp	Exon 5	Yes	None
17:45846707	T>C	rs12373123	24.80	*SPPL2C*	Missense variant	NP_787078.2: p.Ser601Pro	Exon 1	Yes	None
17:45846288	G>C	rs12185233	23.50	*SPPL2C*	Missense variant	NP_787078.2: p.Arg461Pro	Exon 1	Yes	None
17:45825139	C>T	rs4341787	21.70	*CRHR1,* *LINC02210-CRHR1*	Intron variant, Intron variant	N.A	Intron 4, Intron 6	N.A	EH38E1866300—distal enhancer-like signature

Abbreviations: CADD = combined annotation-dependent depletion, cCREs = candidate cis-regulatory elements; N.A = not applicable.

**Table 3 genes-15-00046-t003:** Risk analysis of rs17651549 in Ashkenazi Jews.

Group		Number of Alternate Alleles (AFs)	Number of Alternate Alleles/Total Alleles in AJ Controls (AF) ^a^	Allelic Odds Ratio ^b^ (95% Confidential Interval, *p*-Value)
**AJ-PD patients (n = 1200)**		515 (0.214)	1259/4912 (0.256)	0.793 (0.705–0.891, **0.0001**)
**Stratified analysis**	***GBA1*-PD (n = 235)**	99 (0.211)		0.774 (0.615–0.975, **0.030**)
	***LRRK2*-PD (n = 145)**	56 (0.193)		0.694 (0.515–0.936, **0.017**)
	**NC-PD (n = 787)**	345 (0.219)		0.815 (0.712–0.933, **0.003**)

Note: **^a^** in gnomAD V2.1.1 AJ-non-neuro cases; ^b^ compared to gnomAD V2.1.1 AJ-non-neuro cases. AF = allele frequency; AJ = Ashkenazi Jews; PD = Parkinson’s disease; *GBA1*-PD = PD patients who carry one or more of the ten AJ-*GBA1* mutations and do not carry *LRRK2*-G2019S or *SMPD1*-L302P; *LRRK2*-PD = PD *LRRK2*-G2019S carriers who do not carry *GBA1* mutations or *SMPD1*-L302P; NC-PD = PD non-carriers of any of the ten *GBA1* mutations, the *LRRK2-*G2019S mutation, or the *SMPD1*-L302P mutation; significant *p*-values are in bold.

## Data Availability

The research study protocol and the informed consent form that were approved by the Institutional and National Supreme Helsinki Committees do not allow us to share patients’ information. However, if a specific request is made by a qualified investigator, anonymized data will be shared after approval by our IRB committee, and after an ethical agreement, MTA, is signed between the two institutions.

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
