# Peer review of "MAPT Locus in Parkinson’s Disease Patients of Ashkenazi Origin: A Stratified Analysis"

_genes, 2023, doi:10.3390/genes15010046_

Round 1

Reviewer 1 Report

Comments and Suggestions for Authors

Shani et al present an interesting work, However, there are still many concerns that need to be further clarified. Hope to see their revision.

1. Miami plot of carriers and non-carriers of MAPT haplotypes of selected variants that need to provide

2. For significant loci in MAPT stratified GWAS, OR (95%, Cl) and MAF need to provide

3. Allele-specific interaction between the MAPT tagging SNP and new loci must be analyzed further

4. As the title revealed MAPT Locus in Parkinson’s Disease Patients of Ashkenazi Origin, rare variant analysis needs to show an association of the genes with Parkinson’s disease

5. Also, since almost all variants seemed to be common, whether this is more frequent in the population, please indicate this on other people from Asia, America, Uropian... If they are more seen from the Ashkenazi Origin, please provide patient clinical links to each variant and compare available literature as well to make their finding outstanding

Author Response

Please see the attachment, thanks.

Reviewer 2 Report

Comments and Suggestions for Authors

Comments to the Authors

Major comments

Is there sexodimorphic differences in terms of mutations showed in table-1 between males and females (40 % of patients) for these evaluated  mutations (Carriers of GBA1, LRRK2-G2019S , SMPD1-L302P mutatiosn and patients without these mutations (Non-Carriers)? Is there differences between males and females with PD for the mutations?

-Improve the quality of figure with Schematic representation of the MAPT locus and the linkage disequilibrium (LD) region in Ashkenazi Jews (AJ).

-These authors suggest that recent research demonstrated that LRRC37A/2 is not only involved in pro-inflammatory pathways relevant to astroglial function but also present in substantia nigra astrocytes, where it co-localizes with α-synuclein. This observation may suggest a potential role for LRRC37A/2 in PD pathology [42], and a possible involvement in the effect of H2 on PD. This is strange for me. Please, shall you clarify the real contribution of astroglisis and microgliosis with inflammatory overproduction of cytokines and chemokines and the contribution of LRRC37A/2 in PD. Thanks¡

-Although this study have the limitation that generated by short-reads sequencing technique resulted in two low-coverage regions adjacent to the inversión, understanding its impact on PD protection, they demonstrated a protective effect across all subgroups. In fact, the protective efect of H2 results from different levels of missense mutations in relevant genes, structural variations, epigenetic modifications, and altered expresión of certain RNAs. Collectively, these effects may operate independently or may exhibit synergistic   but suggest a protective effect of MAPT-locus in PD. Please, shall you add more details in the discusión about these mutations and the protective effect of MAPT-locus (Tau) in terms of Tau hyperphosphorilation and the associations with different mutations in rodents models of PD or PD patients with different degress of parkinson diseases (early or advanced PD porgression).

-Please, shall you add additiong information about the exclusión severe GBA1 mutation carriers and compound heterozygotes from this analysis (n=67)

Minnor comments

-Shall you explain the relevance of these H2 haplotypes in european popublation from Polony take into account that are less prevalent that H1 haplptype in Europe and it is more oftein in Africa or America? What does the relevance of these H2 haplotype in terms of Parkinson disease pathology P(PD)?

-This variant showed an association 5 with PD (allelic odds ratio (OR)=0.793, CI=0.705-0.891, p=0.0001, Table3), which was previously reported. However, there was absence of significant association under the H2/H2 recessive 9 model (recessive OR=0.772, CI=0.564-1.056, p=0.106), suggesting a dominant effect of the 10 H2 allele in PD protection. Shall you explain why there was a not signifficant association in this case?

-Thus, what does relathinship exist between of Ashkenazi Origin H2 haplotypes and the progression of Parkinson disease in this population?

-Explain why you assess the effect of the complete H2 haplotype by using three SNVs in LD: rs4528616 at the proximal end, rs62071573 at the distal end, and rs17651549 located at the center of the LD región for a general audience (non expert in Genetics)

-Take into account that approximately nine coding genes reside within the locus, among them is the microtubule-associated protein tau (MAPT) gene, shall you explain the pathological clnical findings in the context of Parkinson disease and the posible Tau deposition in dendrtites from neurons MAPT positive?

-Shall you explain how have you conducted the comprehensive characterization of haplotype-divergent SNVs, predicted to be pathogenic by in-silico analysis tolos? Please, include more details about this thecnic. Thanks¡

-Also, explain how all missense variants were predicted to reside within Evolutionarily Constrained Regions (ECR) by Aminode [25].

However, H2 is not associated with Age at Motor Symptoms Onset 38 Stratified linear regression analysis under a dominant model of H2 revealed that 39 rs17651549 is not associated with AMSO in GBA1-PD (β: 0.770, p= 0.615), LRRK2-PD (β: 40 0.028, p=0.944) and NC-PD (β: 0.056, p= 0.291). Explain, the reason by which these associated are lacked in aging and the possible relevance in Parkinson disease at different stages of this pathology.

These authors indicated in the discussion that both KANSL1 and lysine acetyltransferase 8 (KAT8) have been implicated in PINK1-dependent mitophagy [34], a cellular process that promote degradation of dysfunctional mitochondria, which is associated with PD  [35]. Shall you explain the possible associated between these mutations and H2 haplorypes in terms of Tau hiperphosphorilation. It is possible this association with the WGS mutation?

In addition, the  H2 haplotype is protective in PD; it is possible to suggest that the presence of this additional novel KANSL1 transcript in H2  carriers may contribute to the observed protective effect in PD? What does the association between these KANSL1 transcript in H2  carriers at different degrees of PD diseases (early or advanced) with the possible p-Tau hyperphosphorilation in dopaminergic neurons in terms of differential regulation by autophagic proteins in PD patients with advanced pathology.

My Decision in Minnor Revision

Thanks¡

Comments on the Quality of English Language

Moderate english revision is required.

Author Response

Please see the attachment, thanks.

Round 2

Reviewer 1 Report

Comments and Suggestions for Authors

All concern is addressed accordingly. No further comments.